# Engineering Non-Human RNA Viruses for Cancer Therapy

**DOI:** 10.3390/vaccines11101617

**Published:** 2023-10-20

**Authors:** Vicent Tur-Planells, Adolfo García-Sastre, Sara Cuadrado-Castano, Estanislao Nistal-Villan

**Affiliations:** 1Microbiology Section, Department of Pharmaceutical Science and Health, Universidad San Pablo-CEU, CEU Universities, Urbanización Montepríncipe, 28668 Boadilla del Monte, Spain; v.tur@usp.ceu.es; 2Department of Microbiology, Icahn School of Medicine at Mount Sinai, New York, NY 10029, USA; adolfo.garcia-sastre@mssm.edu; 3Department of Medicine, Icahn School of Medicine at Mount Sinai, New York, NY 10029, USA; 4Global Health and Emerging Pathogens Institute, Icahn School of Medicine at Mount Sinai, New York, NY 10029, USA; 5The Tisch Cancer Institute, Icahn School of Medicine at Mount Sinai, New York, NY 10029, USA; 6Department of Pathology, Molecular and Cell-Based Medicine, Icahn School of Medicine at Mount Sinai, New York, NY 10029, USA; 7Marc and Jennifer Lipschultz Precision Immunology Institute, Icahn School of Medicine at Mount Sinai, New York, NY 10029, USA; 8Icahn Genomics Institute (IGI), Icahn School of Medicine at Mount Sinai, New York, NY 10029, USA; 9Departamento de Ciencias Médicas Básicas, Instituto de Medicina Molecular Aplicada (IMMA) Nemesio Díez, Facultad de Medicina, Universidad San Pablo-CEU, CEU Universities, 28668 Boadilla del Monte, Spain

**Keywords:** oncolytic virus, immunotherapy, cancer vaccines, viral vectors, reverse genetics, virotherapy

## Abstract

Alongside the development and progress in cancer immunotherapy, research in oncolytic viruses (OVs) continues advancing novel treatment strategies to the clinic. With almost 50 clinical trials carried out over the last decade, the opportunities for intervention using OVs are expanding beyond the old-fashioned concept of “lytic killers”, with promising breakthrough therapeutic strategies focused on leveraging the immunostimulatory potential of different viral platforms. This review presents an overview of non-human-adapted RNA viruses engineered for cancer therapy. Moreover, we describe the diverse strategies employed to manipulate the genomes of these viruses to optimize their therapeutic capabilities. By focusing on different aspects of this particular group of viruses, we describe the insights into the promising advancements in the field of virotherapy and its potential to revolutionize cancer treatment.

## 1. Introduction

Cancer management is undergoing a profound advance in both diagnostics and therapeutics. Breakthrough immunotherapeutic interventions such as checkpoint blockade immunotherapy, CAR-T cells, and cancer vaccines set a new generation of therapies to target the unique molecular features of each patient’s cancer. Immunostimulatory targeted cancer therapies harness the immune system’s potential to generate enduring anti-tumoral responses and to establish immunological memory against cancer cells, contributing to improved outcomes of conventional standards of care, while minimizing their off-target toxicities. Work on the development of novel immunotherapies focuses on both prevention (I), driven by prophylactic vaccines, and treatment approaches (II), usually involving combination therapies with standard of care [1,2]. The first one aims to induce specific anti-tumoral memory in healthy people, preventing potential tumors. The latter is applied to cancer patients to boost their immune system against tumoral cells by creating or reinforcing a tumor antigen-specific response, which is of special relevance in cancer patients, who are usually heavily immunocompromised [3].

***Cancer Virotherapy***: The use of viruses as cancer immunotherapeutics is a rising approach towards the generation of long-lasting anti-tumor responses [4]. The selectivity of OVs for tumor cells relies vastly on the same cellular alterations acquired during malignancy progression that ultimately facilitate immune evasion [5]. In some instances, these defects hamper the cancer cell’s capacity to build up an effective antiviral response [6,7], thus increasing the susceptibility of the cell to support viral replication. Detection of viral replication will trigger a cellular response to counteract the progression of the infection by activating Type-I interferon responses, stimulating cytokines release, and switching on different cell-death programs [4,8]. As a whole, this critical direct interaction between the OV and cancer cells will conditionate the remodeling of the tumor immune microenvironment (TIME), setting up the conditions for the innate and adaptive immune branches to engage in the generation of a targeted anti-tumor response, ultimately led by tumor-specific cytotoxic CD8+ T cells [9,10] (Figure 1).

*Engineering viruses for cancer therapy*: The establishment and optimization of reverse-genetics rescue systems, originally developed to understand the biology and pathogenicity of different viruses, have been crucial to advancing OVs into the clinic [11,12]. Reverse genetics enables precise manipulation of the viral genome, which has been instrumental in the design of engineered DNA and RNA OVs (eOVs), with improved tumor specificity, safety, and therapeutic efficacy [13]. Multiple molecular biology techniques are integral to reverse genetics, with their choice contingent upon the virus’s genome type and structure (Table 1). DNA viruses were the first to be genetically modified using homologous recombination technologies [14]. The successful work by Goff and Berg with SV40 [15] was soon adapted and applied to adenoviruses. For RNA viruses, the focus of this manuscript, the foundation of today’s eOV development lies in the groundbreaking research carried out in 1989 by Racaniello and Baltimore, demonstrating, for the first time, the recovery of an infectious RNA virus, Poliovirus ((+) ssRNA), from a complete cloned complementary DNA copy (cDNA) of the viral genome transfected into a mammalian cell system [16,17]. This pioneering work not only paved the way for the development of the polio vaccine but also set up the basis for today’s plasmid-based rescue systems that allowed the production and retrieval of modified infectious viruses [18]. Shortly after, a second milestone in the eOV field came from the group of Peter Palese at Mount Sinai, who rescued the first engineered replication-competent Influenza A virus (eIAV) able to carry and express a foreign gene [19]. This seminal work gave rise to the exploration of viruses as gene expression vectors for different medical applications, including cancer treatment.

***Advancing eOVs into the clinic***: The landscape of eOVs explored in preclinical and clinical settings has been historically dominated by human DNA viruses. The first engineered OV that obtained marketing approval was Oncorine^®^ (2005, China), a genetically modified human adenovirus 5, named H101 (Ad5; dsDNA), with a deletion on the *E1B* gene. Lack of the E1B protein increases the safety of H101, limiting the replication capacity as well as transformation potential of the virus in normal cells [20]. As of today, Oncorine^®^ has been granted clinical application in combination with chemotherapy for the treatment of nasopharyngeal carcinoma and refractory head and neck carcinoma. The therapeutic capabilities of the H101 platform are still under investigation, with 11 active clinical trials in China. In 2015, the regulatory agency FDA (US Food and Drug Administration) approved T-VEC (Talimogene laherparepvec), an engineered human herpes virus (HSV-1, dsDNA) expressing GM-CSF (Granulocyte-Macrophage Colony Stimulating Factor) for the treatment of recurrent, unresectable melanoma. Soon after, IMLIGYC^®^ was granted approval in Europe, Australia, and Israel [21]. Nowadays, T-VEC continues leading the investigation of eOVs in clinical trials, with more than 53 studies registered worldwide. A second HSV-based therapeutic solution, DELYTAC^®^, was approved in 2021 for primary brain cancer in Japan [22]. The technology, Teserpaturev/G47, entails an HSV-1 viral platform developed by the group of Tomoki Todo. As T-VEC, Teserpaturev/G47 continues to be under investigation to expand its potential applicability. The most recently approved eOV by the FDA (December 2022) is Nadofaragene firadenovec (Adstiladrin, also known as rAd-IFNα/Syn3), an adenoviral 5 vector containing the human IFN-α2b gene for the treatment of high risk, non-muscle-invasive bladder cancer [23].

The ability of some non-human viruses to promote anti-tumor responses has been long under investigation [24]. In this review, we cover the main non-human RNA viral platforms being explored as potential cancer therapeutics, along with the molecular characteristics that make them good candidates for cancer virotherapy (Table 2).

## 2. Non-Human RNA Viral Platforms

As indicated in Figure 3, cancer therapy using non-human-adapted viruses offers several advantages. The development of reverse-genetics systems increases the potential for enhancing the oncolytic and immunostimulatory capabilities of these viruses.

### 2.1. Semliki Forest Virus (SFV)

SFV is an enveloped positive-sense single-stranded RNA virus ((+)ssRNA) from the *Togaviridae* family [34]. Its natural hosts are small rodents and birds, and it is usually transmitted by mosquitoes. The physiopathology associated with SFV is strain-specific and symptoms and outcomes could differ between adults and neonatal mice [34]. Human infections are common, and in some parts of Africa, up to 40% of the population is seropositive for SFV [35]. However, SFV could be considered an anecdotical human pathogen. SFV infection symptomatology is mild, presenting fever, headache, and rash [36]. Only one case of a lethal human infection has been reported, which was linked to potential immunodeficiency and exposure to high amounts of the virus in a laboratory setting [37]. SFV has been explored in different preclinical tumor models, including central nervous system (CNS) malignancies, taking advantage of the natural neurotropism displayed by some SFV strains [25].

***Engineered SFV (eSFV) rescue system***: The genome of SFV (around 11 kb) can be modified using reverse genetics [38]. The replication-competent SFV virus can be rescued from a sequence that contains the genes for the non-structural proteins (nsP) 1 to 4, followed by the structural proteins (C, p62, 6K, and E1) open reading frames (*orfs*) in the same expression vector (Figure 2A). In this strategy, incorporating a foreign sequence or *transgene* takes place at the 3′ end of the genome; to function as a viral gene, the subgenomic promoter 6 (SP6) sequence is coupled to its *orf* [39]. The production of replication-deficient SFV particles is achieved using a two-plasmid rescue strategy: an expression vector that encodes the nsP1-4 and the transgene and a helper plasmid encoding the *orfs* for the structural proteins lacking the packaging signals [39]. These vectors are then used to generate *in vitro* recombinant (+)ssRNA genomes under the control of the SP6 promoter. ***Virus recovery***: The positive polarity RNA is transfected into BHK-21 cells, and the proteins codified in the (+)RNA transcripts are directly translated into the cytoplasm. Among those proteins, the SFV RNA replicase can produce a truncated RNA variant that encodes the entire modified SFV genome. The packaging and release of eSFV viral particles is mainly conditioned by the spike membrane proteins E1 and p62, the precursors of E2 and E3 [40]. Interestingly, different studies have shown that other cell lines, such as CHO, HEK293, and neurons, are also suitable for eSFV production, which adds a valuable advantage for further implementation of eSFV production [41].

***Applications***: eSFVs have been explored as cancer vaccine vectors (I) and as OV therapeutic [42] (II). eSFVs have been engineered to express tumor-associated antigens (TAAs) from different malignancies such as mastocytoma (P815) [43], melanoma (MAGE) [44], and cervical carcinoma (E6 and E7) [45]. Moreover, researchers have also explored the use of eSFV as an OV against glioblastoma: by genetically modifying the genome of the virus with microRNA target sequences, eSFV was able to selectively replicate in tumor cells while leaving normal brain cells unharmed, delaying tumor growth and extending the survival of mice [46]. In several preclinical cancer models of colon and lung carcinoma, eSFV carrying IL-12 has been shown to elicit therapeutic antitumor activity [47,48]. Although there have been numerous preclinical studies assessing the applicability of eSFV vectors, only a limited number of clinical trials have been conducted. Most of these trials have been Phase I studies aimed at testing safety rather than efficacy. As a cancer vaccine vector, a Phase I trial evaluated alphavirus vectors expressing the carcinoembryonic antigen (CEA) in patients with advanced pancreatic cancer [49]. The trial found that intramuscular immunizations with doses ranging from 4 × 10^7^ to 4 × 10^8^ IU every week resulted in clinically relevant CEA-specific T-cell and antibody responses after repeated intramuscular administration. The CEA-specific antibodies were able to mediate antibody dependent cellular toxicity against human colorectal cancer cells, resulting in clinical benefits. Similarly, a Phase I trial, focused on the immunization of castration-resistant metastatic prostate cancer (CRPC) patients, tested the efficacy of propagation-defective alphavirus particles expressing the prostate-specific membrane antigen (PSMA). In this case, the viral vaccine administration was completely safe at 0.36 × 10^8^ IU (maximum dose). However, a weak PSMA humoral response and a null cellular response was found [50]. In 2023, there is a Phase II clinical trial recruiting patients, using non-replicating SFV vectors for an HPV-specific therapeutic vaccine, to assess efficacy determined by pathological regression and HPV-16 E6,7-specific T-cell immune responses [51]. LipoVIL12, a therapeutic approach based on lipid particles containing encapsulated eSFV-IL-12 for intravenous administration, has been tested in the clinic on patients with melanoma and renal carcinoma. Following this Phase I trial, it was determined that the treatment had a favorable safety record and did not result in any toxicity upon repeated administration. As a result of this trial, a protocol for a Phase I/II trial in patients diagnosed with glioblastoma multiforme was published [52]. The proposed plan involved administering LipiVIL12 through intratumoral infusion at doses ranging from 1 × 10^7^ to 1 × 10^9^ viral particles. The goal of this trial was to investigate both systemic and local immune responses, as well as to identify factors that predict responses to the treatment. Presently, this study has yet to be conducted.

### 2.2. Seneca Valley Virus (SVV)

SVV is a non-enveloped single-stranded positive-sense RNA virus ((+)ssRNA) that belongs to the genus Seneca virus within the family *Picornaviridae*. SVV was first sequenced and characterized as a contaminant in PER.C6 human embryonic retinal cells [53]. There have been several picornaviruses isolated from swine showing different clinical symptoms across the USA. It is thought that their natural hosts are pigs and possibly other farm animals. SVV is not a human pathogen, nor has it been related to diseases in mice or rats and it does not appear to be prevalent in the human population, as there are no reports of SVV detection in humans [54].

***Engineered *SVV *(eSVV) rescue system***: The SVV genome has a length of 7.2 kb containing a unique *orf* flanked by a 5′ untranslated region (UTR) and 3′ UTR followed by a poly(A) tail [55]. The *orf* codifies for a single polyprotein that undergoes post-translational modifications by virus-encoded proteases to generate the final protein products (5′-L-VP4-VP2-VP3-VP1-2A-2B-2C-3A-3B-3C-3D-3′) (Figure 2B) [56]. One interesting aspect of picornaviruses, and SVV in particular, is the co-translationally cleaving activity of viral polyprotein into the capsid domain (P1) and replicative domain (P2) proteins by a conserved ribosomal skipping sequence in the 2A peptide [56]. The use of 2A ribosomal skipping sequences has enabled the development of expression systems that generate equal amounts of two proteins from a single *orf* [57]. eSVV can be rescued by reverse genetics by introducing the transgene of interest between the 2A and 2B *orfs* in a full-length cDNA vector plasmid. Due to the small size of the SVV genome, the transgene size is limited to 300 to 400 nucleotides [58]. The full-length cDNA of SVV is synthetized *in vitro* by the avian myeloblastosis virus reverse transcriptase using the viral RNA as a template [59]. For transgene expression, the sequence of interest is introduced between the 2A and 2B coding regions by overlap-extension PCR [59]. This PCR amplifies three regions of the full-length SVV cDNA genome that are fused afterward by another round of amplification with primers covering from the 5′ to the 3′ region previously amplified. Finally, the infectious RNA is produced by *in vitro* transcription using T7 RNA polymerase and transfected into PER.C6 cells, where the recombinant virus is produced.

***eSVV applications***: SVV selectivity for tumoral cells relies on the presence of the protein TEM8 (tumor endothelial marker 8), which is highly expressed at the cell membrane surface on certain tumor cells and functions as the preferred receptor for SVV [60]. SVV has shown anti-tumoral effects in different cancer models with neuroendocrine features [61]. SVV has also shown efficacy and safety in several *in vitro* and *in vivo* models of pediatric cancers, including neuroblastoma, rhabdomyosarcoma, and medulloblastoma [62,63], as well as in other tumoral models such as retinoblastoma [64] and glioma [65]. Importantly, SVV has demonstrated anti-tumor capacity when administered systemically, leading to an extent of survival in an orthotopic medulloblastoma cancer model [63]. eSVV’s reverse genetics have allowed the insertion of exogenous therapeutic payloads that can enhance the oncolytic activity of SVV-001 [66]. In the study conducted by Wencheng et al., an oncolytic SVV-A codifying p16^INK4A^ protein, also known as cell cycle-dependent protein kinase inhibitor 2A (CDKN2A), was rescued and characterized [66]. The results showed that SVA-p16 had significantly stronger antitumor effects than the unmodified SVA virus in the human oligodendroglioma Ishikawa cell line. Although SVV-001 has been tested in several Phase I clinical trials, there is only one, at the moment, with public access to the results (NCT00314925). This trial focused on patients with advanced solid tumors with neuroendocrine features in which patients showed good tolerance to intravenous SVV-001 administration at doses of up to 10^11^ vp/kg [67]. All participants demonstrated viral clearance, which coincided with the presence of antibodies against the virus. Small cell lung carcinoma (SCLC) patients exhibited signs of intratumoral viral replication, with estimated peak viral titers exceeding the administered dosage by over 10^3^-fold. One patient, who presented with chemo-refractory SCLC progression, remained progression-free for 10 months after receiving SVV-001 and survived over 3 years after treatment. Given these results, a randomized double-blind Phase II study was performed to assess the efficacy of SVV-001 in patients with extensive-stage SCLC after platinum-based chemotherapy (NCT01017601) [68]. In this case, administration of SVV did not prove advantageous for individuals suffering from ES-SCLC post-platinum-based chemotherapy. The median progression-free survival (PFS) was 1.7 months for SVV therapy and 1.7 months for placebo groups. Furthermore, the persistence of SVV in the bloodstream, 1–2 weeks after treatment, was linked to a reduced PFS. Patients with detectable levels of the virus on day 7 experienced an overall survival (OS) of 3.1 months versus 9.9 months for those who cleared the virus.

### 2.3. Avian Paramyxoviruses (APMVs)

APMVs, commonly referred to as avian paramyxoviruses or avulaviruses, are enveloped, non-segmented negative-stranded RNA avian viruses ((−) ssRNA) classified within the *Paramyxoviridae* family. APMVs have been isolated from a variety of avian species worldwide, including both domestic and wild birds [69,70]. To date, more than 20 different APMV species have been identified, with different avulaviruses displaying strain-specific and host-dependency associated pathology [70,71]. Avian paramyxovirus serotype-1 (APMV-1), also known as Newcastle disease virus (NDV), is the most extensively characterized member of this group. NDV is a major threat to the poultry industry and is endemic in most countries NDV strains are classified based on their virulence in chickens attending to the fusion protein F cleavage site, which is the primary determinant of pathogenicity in birds: lentogenic strains have a monobasic cleavage site in the F protein, usually associated with low virulence and limited replication capacity; velogenic strains have a polybasic cleavage site recognized by endogenous furin-like proteases, allowing for them to be highly pathogenic [72]; mesogenic strains have intermediate virulence, despite having a polybasic cleavage site [73].

The anti-tumor potential of NDV was first reported in 1965, together with the anti-neoplastic and immune stimulatory properties of NDV [74]. Since then, numerous preclinical and clinical studies have demonstrated the potential of NDV as an oncolytic agent [26,27]. In addition, it has been recently reported that other APMVs have oncolytic properties *in vivo* [75]. APMV-4 elicited the best therapeutic responses in melanoma and colon carcinoma preclinical tumor models [75] compared with other APMVs, including NDV.

***Engineered APMV (eAPMV) rescue system***: The genome of all APMVs conserves the canonical six-gene structure (3′-N-P-M-F-RBP-L-5′) characteristic of the family [76]. In the case of NDV, its genome length ranges between 15,186 and 15,198 nucleotides [77] and can be manipulated by reverse genetics. Recovery of infectious NDV from cloned cDNA is achieved by co-transfecting cells that constitutively express the bacteriophage T7 RNA polymerase (BSRT-7) with four plasmids whose expression is under the control of the T7 promoter: full-length antigenomic cDNA (I) and the N (II), P (III), and L proteins (IV) (Figure 2C). To insert a transgene, NDV gene start (GS) and gene end (GE) control sequences need to flank the recombinant *orf*, which is placed in the 3′ non-coding regions of the NDV genome as an additional transcription unit. It is also necessary to comply with the “rule of six”, meaning that the genome nucleotide length must be an even multiple of six for efficient replication. The reinitiation of transcription at the GS in all genes is not perfect, thus leading the polymerase to start transcribing again from the 3′ end, creating a gradient of mRNAs abundance with high levels of mRNA transcription located at the 3′ end [77]. Due to a polar gradient transcription, foreign genes are expressed more efficiently when placed closer to the 3′ end of the genome. While a foreign gene can be inserted between any two genes of NDV, the optimal site for efficient expression and replication is between the P and M genes [78,79,80]. However, the insertion of a transgene increases NDV’s genome length, which impacts virus replication *in vitro* and *in vivo* [81]. NDV can stably incorporate foreign genes at least 4.5 kb in size [82]. Furthermore, an individual NDV-based vector can effectively express at least three different foreign genes [77].

***Applications***: engineered NDVs (eNDVs) have been extensively explored as vaccine vectors; in this matter, the NDV platform presents several benefits: a robust reverse-genetics system that allows for efficient modification and stable expression of foreign genes (I), production of high virus yields in embryonated chicken eggs (II), no risk of gene exchange and recombination on infected cells (III), and an extraordinary capacity to induce mucosal, humoral, and cellular immunity. Different NDV strains have been used by the poultry industry for vaccination against Newcastle disease. The two main vaccine types are live vaccines (I), usually employing lentogenic and mesogenic strains due to their low virulence and safety, and inactivated vaccines (II), where the virus is inactivated after several passages on formalin or β-propiolactone [83,84]. eNDVs have been designed to serve as vaccine platforms for other infectious diseases such as infectious bronchitis virus (IBV) [85], infectious bursal disease virus (IBDV) [77,86,87], and against H5-subtype avian influenza viruses (AIV) [88], with two vaccines already commercialized in China and Mexico [89,90]. Moreover, NDV has also been explored as a vaccine vector for human Influenza A viruses (H1N1), encoding for HA protein [78] and HIV, and encoding for Gag protein [79]. More recently, an eNDV expressing the Spike protein of SARS-CoV-2 has undergone several clinical trials [91].

NDV has a broad cell tropism since the virus receptors are sialic acids, being able to target different cell types. Furthermore, NDV triggers a potent induction of type I IFN, eliciting a strong antiviral innate immune response. The indirect effects of NDV are also of great interest due to their ability to facilitate tumor regression beyond the virus elimination by either infected tumor cells or the host immune system. These effects include the abscopal effect, which targets non-infected tumors, and the provision of long-term protection against tumor recurrence [92].

The ability to perform reverse genetics can be used to enhance the oncolytic potential of NDV. Malogolovkin et al. have conducted a comprehensive review of various studies on eNDV cancer vaccines, oncolytic NDV as a monotherapy, and combinatorial approaches involving NDV and checkpoint inhibitors [27]. Clinical studies conducted on different cancer patients have revealed that NDV causes minimal side effects, with headache, fatigue, and fever being the most reported [93,94]. So far, we have identified 19 clinical trials that utilize NDV as a cancer treatment, either as a single agent (6 trials) or in combination with other therapies (13 trials), but only two clinical trials were registered (NCT03889275, NCT04613492). Cassel et al. conducted several initial studies using autologous or allogeneic NDV oncolysates to vaccinate patients with resected high-risk melanoma. These studies showed a significant improvement in overall survival compared to historical controls [95,96,97,98,99]. Liang et al. employed a comparable method in their Phase III study on colorectal cancer. They compared adjuvant immunization with NDV-modified autologous cancer cells to resection alone. The results showed an improvement in survival in the vaccine group (7 vs. 4.5 years), which was statistically significant [100]. The last two clinical trials and the only ones registered in clinicaltrials.gov are a combination of an attenuated NDV-carrying GM-CSF (MEDI5395) and IL-12 (MEDI9253) with durvalumab (anti-PD-L1) (NCT03889275; NCT04613492) for Phase I evaluation. Numerous trials have demonstrated the clinical advantages of NDV, but only a portion of patients showed positive responses to the treatment. The differences in the response between patients highlight the need to identify predictors that can aid in the selection of patients who are most likely to benefit from this oncolytic viral therapy. Furthermore, it emphasizes the significance of creating recombinant viruses and combination strategies to overcome resistance.

### 2.4. Vesicular Stomatitis Virus (VSV)

VSV is an enveloped negative-stranded RNA virus (ssRNA(−)) that belongs to the *Rhabdoviridae* family within the *Vesiculovirus* genus. While VSV infection is asymptomatic in humans, animals such as horses, pigs, and cattle can become non-lethally infected with clinical signs [101]. VSV produces vesicular disease, manifested by the formation of lesions on the nasal and oral mucous membranes. VSV displays neurotropism that can lead to encephalitis in experimental animals [102,103].

***Engineered VSV (eVSV) rescue system***: The genome length of VSV is approximately 11,000 nucleotides [29]. At the 3′ end of the VSV genome, there is a leader RNA sequence that spans 47 nucleotides followed by five genes arranged in the following order: nucleocapsid protein (N), phosphoprotein (P), matrix protein (M), attachment glycoprotein (G), and the viral RNA-dependent RNA polymerase (L) (Figure 2D). Finally, the 5′ end is capped off with a *trailer* sequence, 54 nucleotides in length [104]. The *leader* sequence plays a crucial role in polymerase binding and signaling during replication and transcription [105]. On the other hand, the *trailer* sequence contains essential elements for replication and cis-acting signals that facilitate the assembly of newly generated RNA genomes into viral particles [106]. VSV transcription initiates at a single 3′-proximal promoter site and monocistronic mRNAs are generated by sequential transcription of each of the genes [107] in a similar way as has been previously described for the *Paramyxoviridae* family, with transcription units flanked by conserved initiation and termination sequences [108]. The gene closest to the promoter is transcribed at a higher rate compared to the genes located further downstream [109]. As we move toward the downstream genes, the transcription rate decreases gradually. This phenomenon, known as attenuation [110], is observed when the polymerase moves across each gene junction, leading to a decrease of approximately 25 to 30% in the transcription of the downstream gene. It is believed that attenuation occurs due to the dissociation of the polymerase during the process of termination and re-initiation [110].

The VSV genome can accommodate the insertion of foreign genes: a full-length cDNA clone of VSV can be assembled de novo from cDNA sequences of each of the VSV gene, including the cloned transgene, using standard cloning techniques [111]. In addition to the cDNA sequences of VSV, the rescue plasmids contain the bacteriophage T7 promoter at the 3′ end and a copy of the self-cleaving HDV ribozyme sequence at the 5′ end. The full-length cDNA plasmid is transfected together with the N, P, and L support plasmids into BHK-21 cells infected with recombinant vaccinia virus (VV) expressing the T7 RNA polymerase. Interestingly, some studies have analyzed the best position for the insertion of a transgene [112]: depending on the site of insertion, viral titers could decrease up to 100-fold, the junction between the N and P genes being the place where the viral fitness was more compromised [112]. The growth curves of viruses with insertions at the P–M, M–G, and G–L junctions show comparable replication kinetics and viral titters when compared with the wild-type virus. VSV has demonstrated preferential replication and oncolysis in cancer cells [113,114]. Recent studies have shown the cytolytic activity of VSV in cancer cells such as malignant glioma, melanoma, hepatocellular carcinoma, breast adenocarcinoma, specific types of leukemia, and tumors associated with prostate cancer [113,115,116,117,118,119,120,121]. VSV has several characteristics that make it a promising OV, including the absence of pre-existing immunity in humans, a small and easily manipulated genome allowing the insertion of many transgenes, cytoplasmic replication without the risk of genome integration into the host cell, viral genome replication independent from the cell cycle, and rapid growth to high titers in a wide range of cell lines, allowing a large-scale virus production platform [122].

One of the unique features of VSV is its pantropic cell target, as it utilizes ubiquitously expressed cell-surface molecules such as the low-density lipoprotein receptor, phosphatidylserine, sialoglycolipids, and heparan sulfate for cell attachment throughout its virion surface G protein [123]. Interestingly, VSV can be pseudotyped since the G protein can be modified by replacing the whole region facing outside of the virion for heterologous glycoproteins from other viruses, transforming its broad tropism to a certain subset of tumoral cells. For example, the G protein region responsible for receptor binding can be replaced by a single-chain antibody (scFv) to target different overexpressed epitopes in certain tumor cells [124]. Therefore, depending on the needs of the therapy and cancer model, VSV can be modified to narrow the tropism, or let the VSV-G protein maintain the broad tropism and modify the genome to insert antigens or additional surface proteins (normally between the G and L genes) to be used as potential cancer vaccines [125]. Finally, many clinical trials have used VSV as an anti-cancer vaccine adjuvant or as an oncolytic virus [126,127]. One of them (NCT03017820) shows complete remissions with a recombinant VSV expressing IFNβ (for conferring specificity and enhancing the anti-tumor response) and a sodium iodide symporter (NIS), for allowing uptake of radiotracers, exhibiting potent viral replication, along with an increase in tumor-reactive T cells [127].

### 2.5. Reoviruses

The genus Orthoreovirus (Order Reovirales, family *Spinareoviridae*) includes non-enveloped, segmented double-stranded RNA (dsRNA) viruses, commonly referred to as reoviruses. Virions contain 10 linear dsRNA molecules [128,129]: three large (L), three medium (M), and four small (S) genomic segments [130]. A single viral protein is encoded by each genomic segment, except for the S1 segment, which encodes two proteins. Each reovirus positive-sense RNA has a 7-methylguanosine cap at the 5′ end, but the 3′ termini are not polyadenylated [131]. Some orthoreoviruses are associated with infectious diseases in humans, such as respiratory tract disorders and gastroenteritis. Birds and mammals have been found to harbor reoviruses, and in some cases these have been related to respiratory and diarrheal disease [132,133]. They are categorized based on their distinct characteristics, including their host ranges and ability to create syncytia that determine two subcategories: fusogenic and non-fusogenic reoviruses [134]. Syncytia formation is not a determining factor in the reovirus life cycle [135].

***Engineered Reovirus rescue system***: There have been reports of plasmid-based reverse-genetics systems for mammalian orthoreovirus (MRV), which have a genome length of approximately 23,000 Kbp [136]. This reverse-genetics system is based on single plasmids encoding each of the ten reovirus genomic segments that are cloned downstream of a bacteriophage T7 RNA polymerase promoter and a sequence of a hepatitis delta virus (HDV) ribozyme immediately downstream of the 3′ end [136] (Figure 2E). The T7 polymerase in the first-generation reovirus plasmid-based reverse-genetics system was supplied by the recombinant vaccinia virus strain DIs (rDIs-T7 pol) [137,138]. In this rescue system, L929 cells are transfected with plasmids encoding each of the ten reovirus gene segments after being infected with rDIs to recover the virus from the plasmids. A second-generation system uses BHK-T7 cells that persistently express T7 RNA polymerase to boost rescue effectiveness. The second-generation system also uses plasmids that encode multiple reovirus gene segments to enhance rescue efficiency. The use of baby hamster kidney cells that express T7 RNA polymerase and fewer plasmids increased the effectiveness of viral rescue, cut down on the amount of time needed to incubate the infectious virus before recovery, and removed any potential biosafety issues with the use of a recombinant vaccinia virus. Reverse-genetics systems have been developed for two prototype reovirus strains: type 1 Lang (T1L) and type 3 Dearing (T3D). Recently, there have been improvements in the rescue of MRV by expressing helper proteins (FAST and VV capping enzymes) in trans in BHK-T7 [139], which produced an increase in the efficiency of the viral rescue. For the introduction of foreign genes, the foreign gene sequence has to be placed after a viral gene segment separated by a P2A sequence. In one study, an oncolytic mammalian orthoreovirus (T3D) was armed with GM-CSF in the S1-segment [140], being stable after several passages.

*Applications*: Reovirus oncolysis primarily operates through apoptosis induction. Infected cells often display features of apoptotic signaling such as the discharge of interferon (IFN) and the stimulation of NF-κB, triggering the expression of antiviral and proinflammatory genes. Reoviruses replicate in the cytoplasm; thus, in healthy cells, new copies of dsRNA will activate protein kinase RNA-activated (PKR), RIG-I, or MDA5, triggering an antiviral response and halting viral replication. Remarkably, in more than 30% of human tumors, there are *Ras* gene mutations [141], and in *Ras*-transformed cancer cells, the PKR pathway is blocked. Therefore, reoviruses preferentially target tumor cells with the oncogenic Ras S pathway, which can be caused by mutations in *KRas*, *BRaf*, and *Egfr* genes [142].

To date, three species of orthoreovirus, including mammalian orthoreovirus (MRV), avian orthoreovirus (ARV), and pteropine orthoreovirus (PRV), have demonstrated oncolytic capacity. MRV type 3 Dearing (Pelareorep), registered as Reolysin^®^ [143], is the only reovirus that has undergone multiple clinical trials, with promising results in Phase I/II (NCT00651157, NCT01533194, NCT01622543) studies. Pelareorep has been proven safe while inducing modest responses against metastatic melanoma [143]. A Phase III clinical trial (NCT01166542) using Reolysin^®^ in combination with paclitaxel and carboplatin versus chemotherapy alone in platinum-refractory head and neck has been recently completed, although presently the results are not yet available. ARV and PRV viruses have shown oncolytic activity against hepatocellular carcinoma and nasopharyngeal carcinoma cell lines, respectively [141]. ARVs infect various avian species, including turkeys, geese, pheasants, and other wild bird species [144]. In one study conducted by Kozak et al. [145], the avian reovirus pathobiology 1 (ARV-PB1])was isolated and tested against hepatocellular carcinoma cells (HCC). ARV-PB1 was able to replicate efficiently and induce a strong cytopathic effect through syncytia formation. ARV-PB1 infection triggered the stimulation of interferon-stimulated genes (ISGs) and induction of apoptosis in HCC while presenting unapparent toxicity in primary hepatocyte cells. The ARS S1133, used as a vaccine in broiler chickens to prevent viral arthritis, has been recently reported to display oncolytic potential *in vitro* and safe applicability *in vivo*, as an intramuscular and oral vaccine in a Kunming mice model. In this study, mice did not undergo weight loss or show any clinical symptoms. Detection of ARV S1133 RNA in multiple murine organs—heart, liver, spleen, lung, and kidney—did not correlate with pathological damage [144].

### 2.6. Murine Leukemia Virus (MLV)

MLV is a type VI retrovirus belonging to the gammaretroviral genus of the *Retroviridae* family. MLV is an enveloped, diploid positive-stranded RNA virus that replicates through a DNA intermediate via reverse transcription [146]. Its genome of 8.3 Kb in size carries three viral genes: *gag* (I), which encodes the capsid, nucleocapsid, and p6 proteins; *pol* (II), which encodes the enzymes reverse transcriptase, integrase, and the viral protease, which is used in the processing of viral proteins; and *env* (III), which contains the sequences that encode the envelope glycoprotein with receptor binding function. MLV was first isolated in mice, where it has been shown to be associated with different pathologies, including T-cell lymphoma, erythroleukemia, immunosuppression, and neurological disorders [147]. MLV strains could be further subdivided attending to their host range and tropism specificity into four different groups: ecotropic (I), xenotropic (II), polytropic (III), and amphotropic (IV) viruses. Ecotropic MLV strains are only able to infect cells of mice and rats, whereas xenotropic viruses can infect cells from other species but display an inability to infect the same cells from which they were isolated (mice and rats). Amphotropic and polytropic viruses present a broader host range and can infect other species as well (including humans) *in vitro*.

*Applications*: Through the process of pseudotyping, the tropism-defining genes of MLV have been used extensively to manipulate host and tissue targeting as vectors in gene therapy [148]. Retroviruses have been widely used in clinical trials as viral vectors for gene delivery or gene modifications in many cancers such as glioblastoma (NCT02414165) or colon carcinoma metastatic to the liver (NCT00035919). Specifically, those based on MLV have been predominantly used in gene therapy trials based on the property to transduce only actively dividing cells: MLV capsid lacks a nuclear localization signal for active transport across an intact nuclear membrane, thus conferring tropism to dividing tumoral cells. MLV vectors mediate the stable expression of transgenes through integration into target cell chromosomes, and they have low immunogenicity compared to other viral vectors [32,149]. Replication-defective retrovirus (RDR) vectors, which have the viral genes gag, pol, and env removed, have been developed as safer gene delivery carriers. However, the low efficiency of gene transfer exhibited by RDR vectors in solid tumors remains a major limitation for cancer gene therapy, despite their use in clinical trials. Therefore, there are replicating retrovirus vectors (RRVs) derived from MLV with a fully viral genome [150,151]. There have been studies with a recombinant MLV vector expressing IL-12 for bladder cancer therapy [152]. The results showed an extent of survival (almost 20% of complete remission) and a reduction of tumoral volume. Another study used a replicating recombinant MLV with two suicide genes for the treatment of breast tumors: yeast cytosine deaminase (yCD) with the prodrug ganciclovir (GCV) and herpes simplex virus thymidine kinase (TK) with 5-fluorocytosine (5-FC) as a prodrug [32]. The results showed that MLV vectors can achieve high transduction efficiency *in vivo* compared to the conventional RDR vectors previously used in clinical trials and RRV replication was highly restricted to the tumor tissue itself. Therefore, exploring reverse-genetics systems with RRVs, specifically MLV, can offer many possibilities for introducing therapeutic foreign genes with a high transduction efficiency.

The RRVs rescue system is slightly different from the RDR vectors. In RDR vectors, the genes gag, pol, and env are supplied by plasmid transfection in trans in 293T cells while the foreign gene is placed along with the corresponding regulatory flanking sequences (LTR and Ψ) in a separated plasmid. Thus, the RRV rescue system differs from RRVs as RRVs need to be replication competent. RRVs contain the full-length cDNA retroviral genome with all the genes, and the foreign gene is placed after an internal ribosome entry site (IRES) sequence downstream of the gen env stop codon and upstream of the 3′ untranslated region (U3) regulatory sequence of the retroviral genome [32,150,151] (Figure 2F).

RRVs such as MLV may offer a promising experimental therapeutic option as an OV, as long as the therapeutic benefits can outweigh the potential risks. These viruses can efficiently and persistently infect cells *in vivo*, resulting in the expression of transgenes that mediate tumor remission. Moreover, MLV infection can persist in residual cancer cells, which potentially may represent the reservoir for RRVs, allowing the long-term expression of a lethal/therapeutic gene [153,154]. This property could be used *in vivo* to halt tumor growth and relapse and mediate prolonged survival.

## 3. Discussion

The oncological community has gained access to meaningful therapeutic innovations in recent years, but cancer still remains the primary cause of death worldwide. Immunotherapeutic approaches such as checkpoint blockade, CAR-T cells, and personalized vaccines are demonstrating a promising but limited responses in patients, with off-target effects and toxicity remaining the major obstacles to overcome. OV therapy has been proven effective in preclinical studies, with early phase clinical studies demonstrating feasibility, safety, and evidence of antitumor efficacy. Virotherapies that involve the use of DNA viruses, with HSV-, adenovirus-, pox-, and AAV-based strategies, continue to dominate the scenario of clinical trials. However, the use of human-adapted OVs may encounter two main difficulties (Figure 3): First, viruses have general mechanisms to counteract the antiviral response on healthy cells, thus being capable of establishing an active infection in the patient. Second is the impact of pre-existing immunity within the human population. Pre-exposure to common human-adapted viruses may induce an antiviral immune response that can counteract the therapeutic effect of the OV, interfering directly with the capacity of the viral vector to infect the tumoral cells by pre-existing neutralizing antibodies. In this matter, the use of non-human-adapted OVs could be advantageous for virotherapy.

The presence of viral-neutralizing antibodies can also limit their targeting. To avoid the consequences of pre-existing immunity, eOVs can be encapsulated in lipid or polymeric particles and be pseudotyped, substituting the surface proteins [155]. Alternatively, non-human OVs’ lack of pre-existing specific immunity can facilitate virus accessibility and replication [36] (Figure 3). Finally, the usage of OVs that are deficient in evading human immune responses, mainly the IFN response, facilitate halting viral replication by healthy cells (Figure 3), while tumoral cells would still be susceptible to viral replication. Non-human viruses’ safety profile is presumably higher as they should undergo an adaptation into humans to become pathogenic [156] (Figure 3).

The diversity of the OV platforms under investigation in clinical trials, particularly in the past ten years, validates the acknowledgment of OVs as a potentially effective alternative treatment by the broader oncology community. A critical component for the application of different cancer virotherapies in the clinic is the route of administration [157]. The most widely explored is intratumoral administration, where the viral concentration can be more precisely controlled at the tumor site [158]. However, the most practical administration route for clinical application is intravenous delivery, which can also be useful for metastatic tumors as the OV can reach cells in the whole organism through the circulation system [159]. Intravenous delivery could cause several problems regarding the immune clearance of the OV and the possible off-target delivery of the OV to healthy cells. There are also other routes of delivery, such as intraperitoneal, subcutaneous, or intrathecal. These routes are designed for specific tumor locations that facilitate the delivery of the OV [157].

***Future perspectives***: virotherapy with several non-human RNA viruses has been proven to have a better therapeutic effect compared with the standards of care when it is used as a monotherapy or combined with chemotherapy or other immunotherapies. The severe side effects observed in initial clinical studies can be alleviated by virus attenuation using reverse-genetics platforms, allowing the field to progress while maintaining safety and public trust. It is expected that the clinical use of non-human therapeutic viruses will expand shortly, with established legal frameworks and procedures for genetically modified viruses.

Neoadjuvant therapy is a crucial medical approach involving treatment before surgery to shrink tumors. This strategy aims to increase the effectiveness of subsequent surgical procedures. Examples of neoadjuvant therapy include chemotherapy, radiation therapy, and hormone therapy, chosen based on tumor type, size, and patient condition [160]. Neoadjuvant therapy stands apart from adjuvant therapy, which is administered after surgery to eliminate residual cancer cells. Neoadjuvant therapy aligns with induction therapy, where initial treatment enhances later interventions. Neoadjuvant therapy is pivotal in modern oncology, enhancing surgical success by reducing tumor size. Oncolytic virotherapy is an interesting alternative to these current neoadjuvant procedures [161]. The rationale behind neoadjuvant therapy using eOVs is threefold. First, it offers the potential to downsize the tumor, making it more amenable to surgical removal. Second, it can trigger an immune response that targets not only the primary tumor but also any micro-metastases that may be present at the time of treatment. Third, it can potentially enhance the efficacy of subsequent therapies, such us chemotherapy or immune checkpoint blockade, by sensitizing the tumor to these treatments.

Combination therapy involving checkpoint inhibitors merges the use of immunotherapy with treatments such as chemotherapy, targeted therapy, or radiation therapy, looking for a synergy to bolster the overall anti-cancer impact. Combining checkpoint inhibitors with chemotherapy or radiation therapy may amplify immune responses by augmenting the release of tumor antigens recognized by immune cells.

Viruses also can mediate the presentation of neoantigens and co-present tumor-specific antigens that the adaptive immune response can target. Viruses can have different tropisms, such as adeno-associated viruses (AAVs), Newcastle disease virus (NDV), or influenza viruses, allowing a certain specificity in the vaccine immunotherapy depending on the OV selected for the tumor model [162]. This response is associated with the stimulation of a wide set of pattern recognition receptors involved in the antiviral response. The combination of these stimuli together with classic treatments is an interesting and promising approach to improve the current state-of-the-art in specific treatments.

## Figures and Tables

**Figure 1 vaccines-11-01617-f001:**
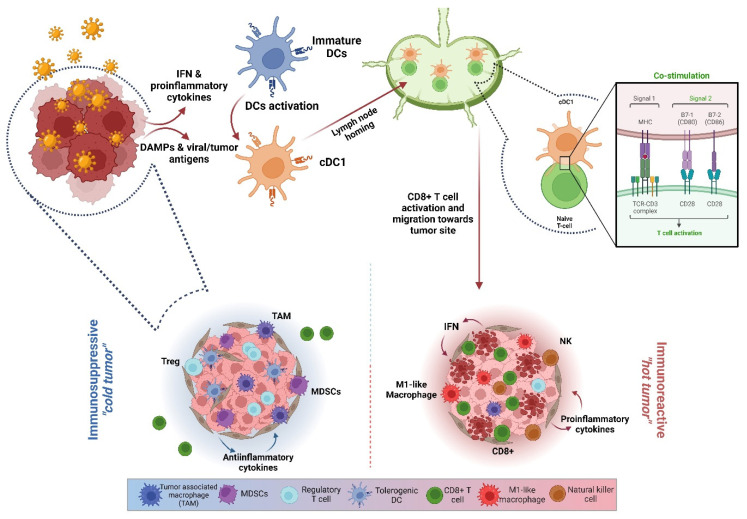
**OV-mediated anti-tumoral immune-response cycle**. Tumor progression is supported by an immunosuppressive TIME, enriched on myeloid-derived suppressor cells (MDSCs), regulatory T cells (Tregs), and tumor-associated macrophages (TAMs). OVs drive tumor immune remodeling: infection of tumor cells triggers the release of proinflammatory cytokines and IFN-related signals that, together with DAMPs and the exposure of viral and tumor-associated antigens by infected cells, stimulate the recruitment and activation of dendritic cells (DCs). Activated DCs then migrate to lymph nodes where they can prime naïve T cells (CD8+ and CD4+). Matured and activated tumor-specific cytotoxic T cells and infiltrated natural killer (NK) cells are the major contributors to tumor debulking.

**Figure 2 vaccines-11-01617-f002:**
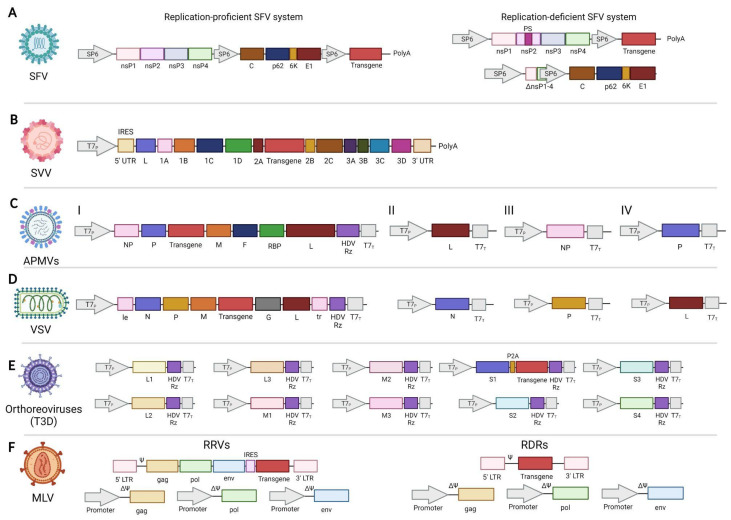
**Reverse-genetics platforms of main non-human RNA viruses used in cancer immunotherapy**. (**A**) Modified Semliki Forest Virus’ (SFV) vectors can be designed for the production of replication-proficient (**right**) or -deficient (**left**) viruses, depending on whether the sequences encoding the structural proteins are present in the same expression plasmid as the non-structural proteins (nsP) (**right**) or in *trans* (**left**). Cloning and expression of a foreign sequence requires the addition of the SFV’ 26S subgenomic promoter. (**B**) Seneca Valley Virus (SVV)-engineered vectors are replication proficient, and the transgene is placed between the 2A and 2B *orfs*. (**C**) Modified avian paramyxoviruses (APMVs) infectious particles can be recovered by employing a four-plasmid (**I**,**II**,**III**,**IV**) rescue strategy. A foreign gene is flanked by a gene start (GS) and gene end (GE) control sequences to act as an additional viral transcription unit; the preferred cloning site for transgene expression is between the P and M genes. (**D**) Vesicular Stomatitis Virus (VSV) replication-proficient viruses are recovered from full-length cDNA expression plasmids. VSV vectors can be pseudotyped by replacing the G gene with a different receptor binding protein sequence. Transgene expression requires additional GS and GE sequences and can be inserted between any two transcription units. (**E**) Engineered mammalian orthroreoviruses (Type 3 Dearing) are replication proficient; a transgene can be expressed after any transcription unit and needs to be flanked by P2A and HDVRz sequences. (**F**) Production of replication-competent (RRVs) Murine Leukemia Viruses (MLVs) requires a full-length cDNA expression plasmid, where a transgene is cloned upstream of the genomic 3′ end; in replication-defective retroviruse (RDR) MLVs vectors, a transgene incorporating the packaging sequences is expressed by itself in *trans*. **Abbreviations**: SP6 = bacteriophage SP6 RNA polymerase promoter; T7_P_ = bacteriophage T7 RNA polymerase promoter; T7_T_ = terminator sequence from bacteriophage T7 RNA polymerase; HDV Rz = hepatitis delta virus ribozyme sequence; Ψ = packaging signal sequence; RRVs = replicating retrovirus vectors; RDRs = replication-defective retroviruses; nsP = non-structural proteins; UTR = untranslated terminal repeats; NP = nucleoprotein gene; L = RNA-dependent RNA polymerase gene; M = matrix protein gene; F = fusion protein gene; RBP = receptor binding protein (HN); G = envelope protein gene of VSV; IRES = internal ribosome entry site; LTR = long terminal repeats; gag = gene encoding for the capsid, nucleocapsid, and p6 proteins; pol = gene encoding for reverse transcriptase, integrase, and protease of retroviruses; env = gene encoding the envelope protein of retroviruses.

**Figure 3 vaccines-11-01617-f003:**
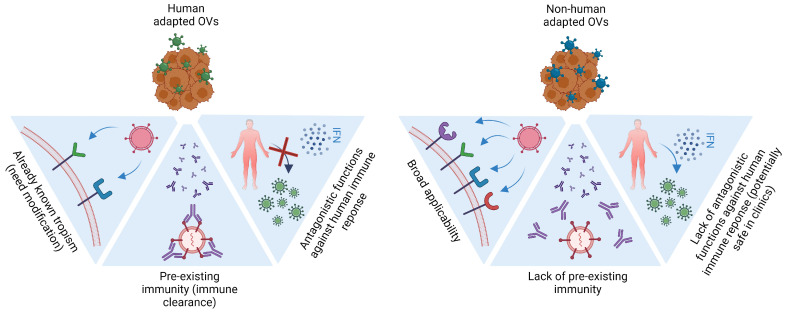
**Non-human OVs in cancer immunotherapy**. ***Broad applicability***: non-human-adapted OVs of tropisms due to species diversity, expanding the possibilities of targeted clinical interventions. ***Pre-existing immunity***: previous exposure to viral agents could be a double-edged sword in cancer virotherapy: for human-adapted OVs, the presence of neutralizing antibodies and virus-specific T-cell responses could conditionate the anti-tumor activity of the viral platform. For non-human viruses, lack of pre-exposure to the viral platform might favor, at least initially, the anti-tumor immune responses by recognizing viral antigens on infected tumor cells. Co-expression of DAMPs, viral and tumor neoantigens in the TIME, further boost a systemic anti-tumor immunity. ***Safe and robust immune stimulators***: lack of human adaptation to counteract the host immune response adds an extra layer of safety to the application of different non-human pathogens in the clinical setting, limiting virus-associated off-targeted pathogenicity in healthy tissues.

**Table 1 vaccines-11-01617-t001:** Molecular biology techniques and steps employed in reverse-genetics rescue systems.

Reverse genetics is a pivotal tool in virology, facilitating the synthesis of viral genomes from cloned DNA fragments, thereby enabling the production of infectious viruses. This approach has revolutionized virus research by providing the means to create tailored viral mutants, elucidate gene functions, and assess viral pathogenicity.
Cloning and manipulation of viral genomes. 1.1 Extraction of viral genomic RNA or DNA. 1.2 Reverse transcription for RNA viruses or PCR amplification for DNA viruses. 1.3 Cloning viral genomic fragments into expression plasmids. 1.4 Site-directed mutagenesis to introduce specific genetic changes. 1.5 ***In vitro*** transcription to generate RNA transcripts or ***in vitro*** DNA synthesis for DNA viruses.
2. Transfection or transduction of host cells. 2.1 Transfection of engineered viral genomes in susceptible cells. 2.2 Delivery method: electroporation, lipofection, viral vectors.
3. Recovery of engineered viruses. 3.1 Co-transfection or co-transduction of host cells with viral genome fragments and helper plasmids expressing essential viral proteins. 3.2 Assembly and packaging of recombinant viral genomes into infectious particles. 3.3 Amplification in permissive cell systems or biological relevant models (embryonated eggs, bioreactors).
4. Characterization of genetically modified viruses. 4.1 Sequencing to confirm the presence of desired genetic modifications. 4.2 Assessing viral growth kinetics, replication, and morphology. 4.3 Validation of transgene expression and function. 4.4 Determining viral pathogenicity through ***in vitro*** and ***in vivo*** assays.

**Table 2 vaccines-11-01617-t002:** Main non-human-adapted RNA viruses for potential virotherapies.

Genus	Species	Viral Genome	Host	Research Stage ^A^	Potential Risk ^B^	Ref. ^C^
Alphavirus	Semliki Forest virus (SFV)	(+) ssRNA	Rodents and birds	2	**Low**: In animals, mainly mice, SFV can produce fever, neurological disorders, and death in severe cases. There are reports of endemic SFV infections in humans in some African regions, causing encephalitis, myalgia, and arthralgia. However, SFV is not known to be directly transmitted between humans as it needs a mosquito vector to be transmitted.	[25]
Orthoavulavirus	Newcastle Disease virus (NDV)	(−) ssRNA	Domestic and wild birds	3	**Low**: no natural NDV infections in the human population. Few incidents of NDV infection in humans occur in laboratory settings with mild symptoms. Economic risk regarding the poultry industry. However, there are vaccines available for chickens.	[26,27]
Senecavirus	Seneca Valley virus (SVV)	(+) ssRNA	Swine and other farm animals. But it is not clear.	3	**High**: no common SVV infection in the human population. As SVV is not a major public health concern and does not have a significant economic impact on the swine industry, there are no vaccines. As a result of the lack of comprehensive data on the vulnerability of other mammalian species and uncertainties surrounding its natural host, SVV can become a high potential environmental risk.	[28]
Vesiculovirus	Vesicular Stomatitis virus (VSV)	(−) ssRNA	Wide host range, including horses, cattle, and swine. Can infect humans also.	3	**Medium**: There are reports of clinical manifestations such as fever and “flu-like” symptoms. The most important vector for VSV transmission is the sand fly or mosquitoes. Exposure to infectious aerosols has resulted in many laboratory acquired infections (between 40 and 46 laboratory associated infections were reported before 1980), although their pathogenicity is relatively mild.	[29]
Orthoreovirus	Mammalian reovirus type 3 Dearing (MRV-T3D) Avian reovirus pathobiology 1 (ARV-PB1)	dsRNA	Mammalian species include humans, rodents, monkeys, and pigs. Primarily birds, including chickens, turkeys, ducks, geese, pheasants, and quails, among others	4 1	**Low**: In both reovirus (MRV and ARV) infections, there have not been reports of clinical symptoms. There is no prevalence of ARV in the human population. The Phase I clinical trial using MRV-T3D showed no dose-limiting toxicity.	[30,31]
Gamma-retrovirus	Murine leukemia virus (MLV)	(+) ssRNA (RT-capacity)	Mice	4	**High**: MLV does not produce natural infection in the human population. However, insertional mutagenesis and activation of proto-oncogene expression have caused T-cell leukemia in some recipients, representing a high potential risk for patients.	[32]

^A^ The stage indicates the phase to which the research has progressed (1: preclinical studies; 2: Phase I clinical studies; 3: Phase II clinical studies; 4: Phase III clinical studies). ^B^ The arbitrary potential risk is scaled as Negligible, Low, Medium, High, and Very High. The scale represents the authors’ estimate of the environmental risks assigned to the clinical use of the viruses based on the aggregate of biological parameters as described [33]. The default class was adjusted if there were factors that positively or negatively affected the risk for the environment or human population. If the available information was found inadequate, the score was increased to provide caution. ^C^ An example of research activities is provided through a single reference.

## Data Availability

Not applicable.

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
