# Peer review of "Engineering Non-Human RNA Viruses for Cancer Therapy"

_vaccines, 2023, doi:10.3390/vaccines11101617_

Round 1

Reviewer 1 Report

In my review of the manuscript “Engineering Non-human RNA viruses for cancer therapy” the author needs to correct the spelling and sentence forming errors, and enrich each section with including more details about the study, in this form this manuscript not acceptable for publication, follow the below comments and solve them, it will help to enrich the content of this manuscript.

Minor comments:

1.      Add a small paragraph related to section 2 then start subsection.

2.      Check the minor grammatical errors.

3.      Check all the references, some are incomplete.

4.      Add reference at line 204.

5.      There are many sentence formation errors, author should proofread it once before submission.

Major comments:

1.      Add the details of a clinical trial at line 209.

2.      Add 2 or 3 detailed figures in the application portion.

3.      Also add one figure in MLV section.

4.      Refer these papers in the introduction https://doi.org/10.3390/vaccines11030507 and 10.3390/ijms232415622

Moderate editing of English language required

Author Response

Enclosed please find the review manuscript entitled Engineering Non-human RNA viruses for cancer therapy.

We are grateful about the response and feed-back received form you over, as well as we appreciate your comments and suggestions that have been used to improve the manuscript quality.

We expect that the changes from the reviewed document to be satisfactory for you.

Comments and Suggestions for Authors

In my review of the manuscript “Engineering Non-human RNA viruses for cancer therapy” the author needs to correct the spelling and sentence forming errors and enrich each section with including more details about the study, in this form this manuscript not acceptable for publication, follow the below comments and solve them, it will help to enrich the content of this manuscript.

Major comments:

  1. Add the details of a clinical trial at line 209.

Thank you for the suggestion, we added more details of the clinical trial at line 228:

“In 2023, there is a phase II clinical trial recruiting patients, using non-replicating SFV vectors for an HPV-specific therapeutic vaccine to assess efficacy determined by pathological regression and HPV-16 E6,7-specific T-cell immune responses (51)”.

  1. Add 2 or 3 detailed figures in the application portion.

We appreciate your suggestion, but we considered that the number of figures was already high, and the scope of this review is slightly different. We think that the explanation of the application is described in each section of each eOV in a detailed way.

  1. Also add one figure in MLV section.

We really appreciate your suggestion, however, please notice that that appears in Figure 2F apart from the explanation of its own section.

Figure 2. Reverse genetics platforms of main non-human RNA viruses used in cancer immunotherapy.               (A) Modified Semliki Forest Virus’ (SFV) vectors can be designed for the production of replication-proficient (right) or deficient (left) viruses, thus depending on whether the sequences encoding the structural proteins are present in the same expression plasmid as the non-structural proteins (nsP) (right) or in trans (left). Cloning and expression of a foreign sequence requires the addition of the SFV’ 26S subgenomic promoter. (B) Seneca Valley Virus (SVV) engineered vectors are replication-proficient and the transgene is placed between the 2A and 2B orfs. (C) Modified avian paramyxoviruses (APMVs) infectious particles can be recovered employing a four-plasmids (I, II, III, IV) rescue strategy. A foreign gene is flanked by a gene start (GS) and gene end (GE) control sequences to act as an additional viral transcription unit; the preferred cloning site for transgene expression is between the P and M genes. (D) Vesicular Stomatitis Virus (VSV) replication-proficient viruses are recovered from full-length cDNA expression-plasmids. VSV vectors can be pseudotyped by replacing the G gene with a different receptor binding protein sequence. Transgene expression requires additional GS and GE sequences and can be inserted between any two transcription units. (E) Engineered mammalian orthroreoviruses (Type 3 Dearing) are replication-proficient; a transgene can be expressed after any transcription unit and need to be flanked by a P2A and HDVRz sequences. (F)Production of replication-competent (RRVs) Murine Leukemia Viruses (MLVs) requires a full-length cDNA expression plasmid where a transgene is cloned upstream of the genomic 3’ end; in replication-defective (RDRs) MLVs vectors, a transgene incorporating the packaging sequences is expressed by itself in trans. Abbreviations: SP6 = bacteriophage SP6 RNA polymerase promoter; T7P = bacteriophage T7 RNA polymerase promoter; T7T = terminator sequence from bacteriophage T7 RNA polymerase; HDV Rz = hepatitis delta virus ribozyme sequence; Ψ = packaging signal sequence; RRVs = replicating retrovirus vectors; RDRs = replication-defective retroviruses; nsP = non-structural proteins; UTR = untranslated terminal repeats; NP = nucleoprotein gene; L = RNA-dependent RNA polymerase gene; M = matrix protein gene; F = fusion protein gene; RBP = receptor binding protein (HN); G = envelope protein gene of VSV; IRES = internal ribosome entry site; LTR = long terminal repeats;  gag = gene encoding for the capsid, nucleocapsid, and p6 proteins; pol = gene encoding for reverse transcriptase, integrase, and protease of retroviruses; env = gene encoding the envelope protein of retroviruses.

  1. Refer these papers in the introduction https://doi.org/10.3390/vaccines11030507 and 10.3390/ijms232415622.

Thank you for your suggestion, we have introduced the second reference you suggest as reference 19 in the manuscript. With respect to the first reference suggestion, it escapes the scope of this minireview and we have not considered it.

Minor comments:

  1. Add a small paragraph related to section 2 then start subsection.

A small paragraph related to section 2 (Non-human RNA viral platforms) was added:

“As previously indicated in Figure 3, cancer therapy using non-human adapted viruses offers several advantages. The development of reverse genetics systems increases the potential for enhancing the oncolytic and immunostimulatory capabilities of these viruses.”

  1. Check the minor grammatical errors.

We have considered your suggestion and checked the grammatical errors we have found to improve the reading.

  1. Check all the references, some are incomplete.

We have included and changed references according to the explanation described in the text.

  1. Add reference at line 204.

The reference to it is located in the previous sentence.

  1. There are many sentence formation errors, author should proofread it once before submission.

We have reviewed all the text and improved grammar and vocabulary in order to facilitate the reading of the review by a professional editor.

Reviewer 2 Report

In this review, the authors systemically summarized and reviewed the critical update of the application of non-human-adapted RNA viruses engineered for cancer therapy. They also describe the diverse strategies employed to manipulate the genomes of these viruses, especially describing the insights into the promising advancements of virotherapy for cancer treatment. This review is well organized and presented, and should be accepted after some minor revision.

1.      It would make audience easier to understand antitumor mechanism if the authors could draw a schematic/cartoon on viruses as cancer immunotherapeutics.

2.      Could you give a brief comparison in the table between human-adapted Ovs-based cancer therapy and non-human-adapted Ovs cancer therapy?

3.      Could you list the general/special methodology adopted to engineer/manipulate the genomes of non-human adapted RNA viruses for potential virotherapies in table 1?

4.      Could the authors give a brief summarization of different administration methods of on-human adapted RNA viruses?

 Minor editing of English language required

Author Response

Dear Reviewer 2,

Enclosed please find the review manuscript entitled Engineering Non-human RNA viruses for cancer therapy.

We are grateful about the response and feed-back received form you over, as well as we appreciate your comments and suggestions that have been used to improve the manuscript quality.

We expect that the changes from the reviewed document to be satisfactory for you.

In this review, the authors systemically summarized and reviewed the critical update of the application of non-human-adapted RNA viruses engineered for cancer therapy. They also describe the diverse strategies employed to manipulate the genomes of these viruses, especially describing the insights into the promising advancements of virotherapy for cancer treatment. This review is well organized and presented, and should be accepted after some minor revision.

Thank you. We appreciate your comments.

  1. It would make audience easier to understand antitumor mechanism if the authors could draw a schematic/cartoon on viruses as cancer immunotherapeutics.

We appreciate your suggestion and we decided to generate and include a Figure in the manuscript under Figure 1.

Figure 1.  OV-mediated anti-tumoral immune response cycle. Tumor progression is supported by an immunosuppressive TIME, enriched on myeloid derived suppressor cells (MDSCs), regulatory T cells (Tregs) and tumor associated macrophages (TAMs). OVs drives tumor immune-remodeling: infection of tumor cells triggers the release of proinflammatory cytokines and IFN-related signals that, together with DAMPs and the exposure of viral and tumor associated antigens by infected cells, stimulate the recruitment and activation of dendritic cells (DCs). Activated DCs migrate then to lymph nodes where can prime naïve T cells (CD8+ and CD4+). Matured and activated tumor-specific cytotoxic T cells and infiltrated natural killer (NK) cells are the major contributors to tumor debulking.

  1. Could you give a brief comparison in the table between human-adapted OVs-based cancer therapy and non-human-adapted Ovs cancer therapy?

We appreciated your suggestion and we considered that this comparison could fit better in a Figure. Therefore, we have generated a new Figure -Figure 3- highlighting:

Figure 3. Non-human OVs in cancer immunotherapy. Broad applicability: non-human adapted OVs of tropisms due to species diversity, expanding the possibilities of targeted clinical interventions. Pre-existing immunity:  previous exposure to viral agents could be a double edge sward in cancer virotherapy: for non-human adapted OVs, the absence of pre-existing neutralizing antibodies or virus-specific T-cell responses in the general population favors the stimulation of anti-tumor immune responses; viral proteins expressed on infected cancer cells contribute to antigen spread and could be recognized as tumor neoantigens, further boosting anti-tumor immunity. Safe and robust immune stimulators: lack of human adaptation to counteract the host immune response adds an extra layer of safety to the application of different non-human pathogens in the clinical setting, limiting virus-associated off-targeted pathogenicity in healthy tissues.

  1. Could you list the general/special methodology adopted to engineer/manipulate the genomes of non-human adapted RNA viruses for potential virotherapies in table 1?

Thank you for your suggestion, we included Table 1 where the molecular biology techniques and steps employed in reverse genetic rescue systems are explained.

Table 1. Molecular biology techniques and steps employed in reverse genetic rescue systems.

Reverse genetics is a pivotal tool in virology, facilitating the synthesis of viral genomes from cloned DNA fragments, thereby enabling the production of infectious viruses. This approach has revolutionized virus research by providing the means to create tailored viral mutants, elucidate gene functions, and assess viral pathogenicity.  

1.      Cloning and Manipulation of viral genomes 

1.                Extraction of viral genomic RNA or DNA. 

2.                Reverse transcription for RNA viruses or PCR amplification for DNA viruses. 

3.                Cloning viral genomic fragments into expression plasmids. 

4.                Site directed mutagenesis to introduce specific genetic changes. 

5.                In vitro transcription to generate RNA transcripts or in vitro DNA synthesis for DNA viruses. 

2.      Transfection or transduction of host cells. 

2.1  Transfection of engineered viral genomes in susceptible cells. 

2.2  Delivery method: electroporation, lipofection, viral vectors. 

3.      Recovery of engineered viruses 

1.                Co-transfection or co-transduction of host cells with viral genome fragments and helper plasmids expressing essential viral proteins. 

2.                Assembly and packaging of recombinant viral genomes into infectious particles. 

3.                Amplification in permissive cell systems or biological relevant models (embryonated eggs, bioreactors). 

4.      Characterization of genetically modified viruses 

4.1 Sequencing to confirm the presence of desired genetic modifications. 

4.2  Assessing viral growth kinetics, replication, and morphology. 

4.3  Validation of transgene expression and function. 

4.4  Determining viral pathogenicity through in vitro and in vivo assays. 

  1. Could the authors give a brief summarization of different administration methods of on-human adapted RNA viruses?

We included a brief summary of the different administration routes at line 588: “A critical component for the application of different cancer virotherapies in the clinic is the route of administration https://www.ncbi.nlm.nih.gov/pmc/articles/PMC7176816/. The most widely explore is the intratumoral administration,where the viral concentration can be more precisely controlled in the tumor site https://pubmed.ncbi.nlm.nih.gov/30115692/. However, the easiest administration route for clinical application is the intravenous delivery, and can also be useful for metastatic tumors as the OV can reach cells in the whole organism through the circulation system https://pubmed.ncbi.nlm.nih.gov/21674738/. Intravenous delivery could arise several problems at the immune clearance of the OV and the possible off-target delivery of the OV to healthy cells. There are also other routes of delivery such as intraperitoneal, subcutaneous, or intrathecal. These routes are designed for specific tumor locations that facilitate the delivery of the OV https://www.ncbi.nlm.nih.gov/pmc/articles/PMC7176816/.  “

Round 2

Reviewer 1 Report

There are issues of referencing part; kindly check and modify

Minor editing of English language required

Author Response

Dear reviewer:

Thank you for your time. Below we indicate the changes as you have suggested.

Comments and Suggestions for Authors

1.- There are issues of the referencing part; kindly check and modify

We have reformatted the references section and made the critical changes that could be considered problematic (Invalid references). MDPI formatting has also been used this time.

2.- Comments on the Quality of English Language

Minor editing of English language required.

English has been edited by an English editor. We have corrected it accordingly. MDPI has also indicated to us that the editorial counts on editors to double-check the text. We hope this time we can have a better version.